# Analysis of the Expression and Subcellular Distribution of *eEF1A1* and *eEF1A2* mRNAs during Neurodevelopment

**DOI:** 10.3390/cells11121877

**Published:** 2022-06-09

**Authors:** Zoe Wefers, Celia Alecki, Ryan Huang, Suleima Jacob-Tomas, Maria Vera

**Affiliations:** Department of Biochemistry, McGill University, Montreal, QC H3G 1Y6, Canada; zoe.wefers@mail.mcgill.ca (Z.W.); celia.alecki@mail.mcgill.ca (C.A.); ryan.huang1@mail.mcgill.ca (R.H.); suleima.jacob@mail.mcgill.ca (S.J.-T.)

**Keywords:** eEF1A1, eEF1A2, mRNA localization, neurodevelopment, smFISH, imaging analysis

## Abstract

Neurodevelopment is accompanied by a precise change in the expression of the translation elongation factor 1A variants from eEF1A1 to eEF1A2. These are paralogue genes that encode 92% identical proteins in mammals. The switch in the expression of eEF1A variants has been well studied in mouse motor neurons, which solely express eEF1A2 by four weeks of postnatal development. However, changes in the subcellular localization of eEF1A variants during neurodevelopment have not been studied in detail in other neuronal types because antibodies lack perfect specificity, and immunofluorescence has a low sensitivity. In hippocampal neurons, eEF1A is related to synaptic plasticity and memory consolidation, and decreased eEF1A expression is observed in the hippocampus of Alzheimer’s patients. However, the specific variant involved in these functions is unknown. To distinguish eEF1A1 from eEF1A2 expression, we have designed single-molecule fluorescence in-situ hybridization probes to detect either *eEF1A1* or *eEF1A2* mRNAs in cultured primary hippocampal neurons and brain tissues. We have developed a computational framework, ARLIN (analysis of RNA localization in neurons), to analyze and compare the subcellular distribution of *eEF1A1* and *eEF1A2* mRNAs at specific developmental stages and in mature neurons. We found that *eEF1A1* and *eEF1A2* mRNAs differ in expression and subcellular localization over neurodevelopment, and *eEF1A1* mRNAs localize in dendrites and synapses during dendritogenesis and synaptogenesis. Interestingly, mature hippocampal neurons coexpress both variant mRNAs, and *eEF1A1* remains the predominant variant in dendrites.

## 1. Introduction

Humans’ remarkable capacities to learn, consolidate memories, and perform complex tasks are all related to the postnatal extension of brain development. Neurons grow into complex polarized cells with a long axon and elaborate dendritic branches that expand far from the nucleus and soma. Neuronal networks are created through the formation of synapses between axons and dendrites of different neurons. The undifferentiated neuron starts its journey through maturation by forming neurites where growth cones will branch in the direction of the stimulus, seeking synapse formation [1]. Hence, the process of neurodevelopment tackles multiple changes in the architecture and protein composition of neurons that are fundamental to establishing the functional circuits of the mature brain. This complex process requires the compartmentalization of functions through a decentralized gene expression system [2], which relies on mRNA localization [3], tight regulation of local translation [2,4,5], and sustained local protein homeostasis [6]. These events lead to local transcriptomes that are vast and diverse and to significant changes in the expression of almost two thousand proteins during neuronal differentiation [7].

An intriguing gene expression switch during postnatal neuronal development is the change in the eukaryotic translation elongation factor eEF1A variant 1 (*eEF1A1*) to its paralogue gene *eEF1A2*. *eEF1A1* and *eEF1A2* genes encode 92% identical and 98% similar proteins in mammals [8]. Despite this remarkable similitude, their pattern of expression is opposite: eEF1A2 is not expressed during prenatal development, the joint expression of both variants is restricted to the postnatal development of a few specific cell types (mainly neurons and muscle cells (myotubes and cardiomyocytes)), and they become mutually exclusive in differentiated healthy adult cells [9]. The complete switch in the postnatal expression of these paralogue genes was first identified in motor neurons of the wasted mouse, which carries a spontaneous deletion of chromosome 2 where the *eEF1A2* locus is located [10]. The absence of eEF1A2 initiated a neurodegenerative phenotype only at postnatal day 21, when eEF1A2 fully replaced eEF1A1 expression [11,12]. The subsequent inability of motor neurons and muscle cells to synthesize new proteins leads to death on postnatal day 28 [10]. These dramatic and opposed changes in the expression of eEF1A1 and eEF1A2 in mouse motor neurons have prompted the hypothesis that eEF1A2 provides functional and physiological advantages to neurons in general, even though the timing and extent of the switch in the expression from eEF1A1 to eEF1A2 proteins have not been characterized in other neuronal types. In humans, mutations in the *eEF1A2* gene have been demonstrated to cause intellectual disability, autism, seizures, and epilepsy [13,14,15,16], and, thus, the actions of eEF1A2 in human neurons ensure normal neurodevelopment.

Both eEF1A variants are endowed with the capacity to coordinate changes in local protein synthesis and cytoskeleton reorganization in neurons. Therefore, they sustain the functional development of the brain [17,18,19,20]. The canonical role of both eEF1A1 and eEF1A2 in translation elongation is to bring aminoacyl transfer RNAs (tRNAs) to the A site of the 80S ribosome in a GTP-dependent manner [8,11,21,22]. Further, eEF1A was initially identified as an actin-binding protein that can also bundle actin filaments [23] and remodel the actin cytoskeleton [24]. It was later shown that its capacity to bind F-actin is decreased by the presence of aminoacyl tRNAs [25,26,27,28,29]. Adult neurons, and particularly hippocampal neurons, precisely adapt local protein synthesis and remodel the actin cytoskeleton of dendritic spines to support long-lasting synaptic plasticity and memory consolidation [30]. Accordingly, the *eEF1A* mRNA and protein were detected in the axon of the sea slug *Aplysia* [31] and the dendrites of different neuronal types upon synaptic stimulation, where the local synthesis of eEF1A is regulated by the Fragile X mental retardation protein (FMRP) [32,33,34,35]. Downregulation of the eEF1A protein has been found in the hippocampus of brain samples of Alzheimer’s patients, and it correlates with defects in protein synthesis and synaptic plasticity in the Alzheimer’s mouse model Tg2576 [36]. Although most of these studies do not distinguish between eEF1A variants, they strongly ascribe an essential role to eEF1A in hippocampal development and function. Neurons of young mice showed a higher localization of eEF1A in neuronal processes than adults [34], but both *eEF1A1* and *eEF1A2* transcripts had been sorted as being translated in the postsynapses of the CA1 neurons of the adult rat hippocampus by Ribosome profiling [37]. Thus, we predict both eEF1A variants to be expressed in hippocampal neurons, and *eEF1A1* and *eEF1A2* mRNA localization could be a read-out for protein distribution in hippocampal neurons.

In this paper, we have developed an approach to investigate changes in the expression of eEF1A1 and eEF1A2 in hippocampal neurons over development and differentiated hippocampal neurons based on the spatiotemporal regulation of their encoding mRNAs. To this end, we analyzed the expression and subcellular distribution of mRNAs in individual neurons by single-molecule fluorescence in situ hybridization (smFISH) [38,39,40,41,42]. Indeed, smFISH has several advantages over immunofluorescence in studying changes in the expression of eEF1A variants in neurons. Firstly, the nucleotide identity between them is 75% in the coding sequence, and they have different 3′- and 5′- untranslated regions [43], which allowed us to design specific smFISH probes that do not cross-recognize the other eEF1A variant. Secondly, smFISH is a very sensitive and robust technique that detects 90% of the mRNAs in a fixed cell and provides the single-molecule resolution and precise quantifications [38]. On the contrary, more than one protein is required for immunofluorescence to provide a signal over the autofluorescence background of the sample. To extract changes in the subcellular distribution of mRNAs and relate them to specific neuronal compartments from the smFISH data, we have created a computational framework, ARLIN (Analysis of RNA Localization In Neurons) “https://github.com/VeraUgalde/ARLIN (accessed on 20 May 2022)”. ARLIN is a semi-automatic tool to segment neurons into dendritic, somatic, and nuclear compartments that provides compartment-specific mRNA localization analysis. Quantitative techniques to analyze subcellular mRNA localization patterns have been developed for non-neuronal cells. Battich et al. developed a method that used a multivariate feature set quantifying spatial properties of transcripts to analyze localization patterns and their cell-to-cell variability [44]. DypFish improved the quality of analysis by quantitatively evaluating colocalization and clustering patterns of transcripts and how these patterns correlated to the protein products of the transcripts [45]. Most recently, the group of Mueller released FISHQuant v2 https://github.com/fish-quant (accessed on 20 May 2022), which contains highly accurate machine learning models that can identify several subcellular localization patterns from smFISH images [46,47,48]. However, all these methods rely on a predefined set of features, including statistics, such as the distance of each transcript to the cell centroid. Such statistics are not particularly meaningful to neurons because of their extended dendritic compartments. Furthermore, the cluster and colocalization patterns identified by these methods do not consider the general tendency for neurons to have a high mRNA density in somatic compartments. Therefore, ARLIN is a neuron-specific tool that can collect statistics about dendritic distances and analyze the localization patterns of somatic and dendritic transcripts separately. Accordingly, it has enabled us to analyze subtle changes in the localization of *eEF1A1* and *eEF1A2* transcripts as the first step towards understanding the complex regulation of expression of these proteins in neurons.

## 2. Materials and Methods

### 2.1. Mice

WT C57BL/6 black mice were purchased from Charles River Laboratories. All experimental procedures involving mice were conducted in compliance with the Canadian Council on Animal Care guidelines and were approved by the Facility Animal Care Committee (FACC) of McGill University.

### 2.2. Neuronal Cultures

Primary hippocampal neuronal cultures were obtained from postnatal day 0 (P0) WT C57BL/6 black mice pups and prepared as previously described [30]. Briefly, brains are extracted from P0 pups, a mid-sagittal cut is performed to separate the cerebral hemispheres, the midbrain is then removed, and the hippocampus (HC) isolated. All isolated HCs are placed in a falcon tube on ice containing 1× Hanks’s balanced salt solution (HBSS) buffer consisting of 1× HBSS and 1 M HEPES in RNase-free water. Following isolation, HC tissue is dissociated by adding 1× Trypsin to a falcon tube containing HC and incubated at 37 °C for 15 min to isolate neurons. After incubation, trypsin and HBSS are removed, and neurons are washed at room temperature (RT) using fetal bovine serum (FBS) medium containing 5% FBS, 100× GlutaMAX supplement, 50 mg/mL Primocin, and 1× Neurobasal-A medium. FBS medium is then removed, fresh FBS medium added, and HC tissue titrated to further isolate neurons. Isolated neurons are then counted, and ~70,000 neurons are seeded in MatTek^®^ dishes coated with 0.2 mg/mL PDL coating solution containing 2 mg/mL poly-D-lysine (PDL) solution in 10× Boric Acid Buffer (BAB) (50 mM boric acid and 12.5 mM Borax pH adjusted to 8.5). Neurons are cultured in vitro and maintained using NGM medium consisting of 1× B-27 Supplement no serum, 1× GlutaMAX™ Supplement, Primocin, and 1× Neurobasal-A medium until DIVs 1, 3, 5, 9, 14, 21.

### 2.3. smFISH and Immunofluorescence (smFISH-IF)

Stellaris smFISH probes labeled with Quasar 570 (CY5) were purchased from LGC Biosearch Technologies to identify *mus musculus eEF1A1* mRNA: cgcacttatataccgttctc, ggcaaacccgttgcgaaaaa, ttttcacaacacctgcgttc, tcccatttttgctttgaatt, tgttgatgtgagtcttttcc, ggaatctacgtgtccgatta, ttccaccacatttgtagatc, aacttttcgatggttcgctt, tttcagtttgtctaagaccc, caatagtgataccacgctca, tcgaatttccacagggagat, ggcatcaatgatggtcacat, tgatgaagtctctgtgtcct, gatgtgcctgtaatcatgtt, acaatcaggacagcacagtc, caaattcaccaacaccagca, ccaacaatcagctgtttcac, cggtggaatccattttgttg, atctcttctgactgtatggt, acttccttaacgatttcctc, ggttgtagccaattttctta, tggcacaaatgctactgtgt, catgttgtcaccattccaac, accaaggcatattagcactt, tacaatccaaagcttccagc, cttgtcagttggacgagttg, ccaatgcctccaattttata, gttgttacattgactggagc, ggtgcatttcaacagacttg, agagcttcactcaaagcttc, ttctttacattgaagcccac, cgtctaacatctttgaccga, cctggatggttcaggataat, gacaatccagaacaggagcg, ctcagcaaacttgcatgcta, acgacgatcgatcttttctt, cagacttcaggaacttgggg, ccatatcaacaatggcagca, tctcaacacacatgggcttg, agtggagggtagtcagagaa, gtcacgaacagcaaagcgac, ttgtccacagctttgatgac, atttagccttctgagctttc, gcaggtgttaggggtaatat, accactgattaagactgggg, aacagttctgagaccgttct, aggttttacgatgcattgtt, ctgtgacagatttttggtca, and *eEF1A2* mRNA labeled with Quasar 570 (CY5): cgtaatgaggatccgagagg, aggactcagtcagaaggact, caatctcatcatagcgcttc, ccttgaaccatggcatatta, gacgtagagcctcttacaag, gtgagaagtgccagactgaa, cacccaaaaagggtcatagc, cagtcagttttggttctact, cactgaagtcatggcatgtt, ttaagaaccagtactctgcc, cctaggagactgacacaagg, gctctagaagagatgagggt, aggttctgaagatgcactca, tagcatagaggctgagtgta, aaccacctttccaaacacat, attagttcaaacccatcctg, cttgggtatctgtatccaat, ggtctacctgatcaatcttg, cttagttttcccaatcctaa, tcccatctcagtctaatgaa, tagggtaacagggccaatag, attatcaccacaaccctgaa, gaaacccagtgggaagtaga, tttcagctcagatgtatggt, ggaatgtgcaacagttgctg, agaaatcagccttcagtctc, ggtgatacactcttattcca, cacttcaaccctgaaactgt, tgcctcatgaatctttcaga, atatttggccaactcagcaa, caaaagttgccacaaggctc, tgtgcatacgtatgctatct, catgatcacgcaggcatata, gggaatggattttcctttca, tctatccttgtttctgtgat, actgtagccaagacaaaggc, cagaaagtgagattctccct, tgaagaccctttccactaag, agcggtaagggggatcaaag, cagggatgcatctcggaaag, ttcatggctgtgagcttaac, cactgttctgagtacaccaa, cttagcatgaaacagcctca, atgtagagtttcaaagccca, aaatatttgcccttataccc, ctcctttctctgtgataact, agggacatttcttcaaggga, cttgctttcttatagaaccc.

We adapted an smFISH protocol from and described in Eliscovich et al. [42]. Briefly, neurons are fixed for 15 min on ice using 4%PFA in PBSM containing 1XPBS and 5 mM MgCl_2_. Following fixation, neurons are quenched for 10 min on ice using 0.1 M Glycine in PBSM. Neurons are then permeabilized for 15 min on ice using 2 mM Ribonuclease Vanadyl Complex (VRC, Sigma-Aldrich, St. Louis, MO, USA) and 0.1% Triton X-100 in 1XPBS. Following permeabilization, neurons are then incubated with prehybridization solution containing 10% formamide and 2× saline sodium citrate (SSC) for 30 min at RT. Afterwards, neurons are incubated for 3 h at 37 °C with hybridization solution containing 2XSSC, 10% formamide, 1 mg/mL *E. coli* tRNA, 10% dextran sulfate, 0.2 mg/mL ultrapure BSA, 2 mM VRC, 10 U/mL Superase, 125 nM of smFISH probes (eEF1A1 (CY5) and eEF1A2 (CY5), and primary antibodies to identify neuronal structures (dendrites: MAP2; Rabbit polyclonal antibody; 1:500 and post-synaptic structures: PSD95 mouse monoclonal antibody; 1:1000). Following incubation with hybridization solution, neurons were incubated for 1 h at 37 °C with fluorescently labeled secondary goat antibodies against rabbit IgG conjugated with Alexa Flour 750 (Invitrogen, mouse IgG conjugated with Alexa 488 (Invitrogen, Waltham, MA, USA), all diluted at 1:1000 in prehybridization solution. Neurons are then washed twice with 2× SSC for 10 min at RT. To prevent photobleaching, a drop of ProLong™ Gold Antifade Mountant solution containing DAPI (to label nucleus) is placed onto the neurons in the glass bottom of the MatTek^®^ dish, followed by a glass coverslip. Neurons are left overnight at RT protected from light to allow the antifade solution to dry. Following overnight incubation, neurons are stored at 4 °C until imaging acquisition.

### 2.4. RNA Isolation and RT-qPCR

Hippocampal neurons were grown on transwell membrane cell inserts, and total RNA was isolated from soma (up) and neurites (bottom) fractions as previously described [49]. After a wash with PBS, the soma fraction was scrapped from the membrane, placed into a tube, and centrifuged 2 min at 2000 g, resuspended in 400 µL of ice-cold PBS, and 750 µL of Zymo RNA lysis buffer (Zymo Quick RNA miniprep kit, Zymo Research, Irvine, CA, USA) were added to each tube. Membranes were cut from the transwell, put face down in a 6 cm plate containing 750 µL of Zymo RNA lysis buffer, and incubated 15 min on ice while tilting the plate every few minutes. After 15 min, neurites containing solution were transferred into a tube, and RNA isolation from soma and neurites was completed. Twenty-five ng of RNA isolated from soma or neurites were reverse transcribed into cDNA using iScript™ Reverse Transcription Supermix (Bio-rad, Hercules, CA, USA) following manufacturer instructions. For qPCR, cDNA was diluted two-fold in water. PCR was performed in 5 mL reactions consisting of 1 mL of DNA < 2.5 mL PowerUp SYBR Green master mix (ThermoFisher, Waltham, MA, USA) and 0.25 mL of 1 mM of each primer. Standard curves were generated using a log titration of N2A cDNA (50 to 0.05 ng). Data were collected using Viaa7 PCR system with 45 cycles. The standard curve was used to calculate cDNA amounts. Primer sequence for eEF1A1F: ACTTGGGGCCATCTTCCAGC, eEF1A1R: AGTGCTGGCTACGCTCCTGT, eEF1A2F: GCCATAGTAAAGGAACCCCTG, and eEF1A2R: GTGTGTGGAGAGCTTCTCAC.

### 2.5. Tissue smFISH

Paraffin-embedded slides of 6- or 90-week-old male C57BL/6J mice were obtained from the Jackson laboratory. The smFISH to detect eEF1A1 and eEF1A2 mRNAs were completed in 4 mm thick sagittal slides corresponding to section 18 of Allen’s Brain mouse atlas (P56) as they have most of the hippocampal formation (DG, CA1, CA2, CA3). We used a protocol previously described by Annaratone et al. [50]. The specificities for the eEF1A probes were to use them at a final concentration of 125 nm in 15% formamide and hybridized at 37 °C overnight.

### 2.6. Imaging Acquisition

Primary hippocampal neurons were imaged using a custom widefield inverted Nikon Ti-2 widefield microscope equipped with CFI PLAN APO LAMBDA 60× 1.4 NA oil immersion objective lens (Nikon), Spectra X LED light engine with a C-FL DAPI SOLA, C-FL GFP/FITC/Cy2, C-FL DSRed/TRITC/Cy3, and C-FL Cy5 filter sets (Semrock, Rochester, NY, USA) and a custom Cy7 Penta Pass Filter for NIR SpectraX, DAPI-FITC-TRITC-CY5-CY7 bandpass filter, and an Orca-Fusion sCMOS camera (Hamamatsu, Hamamatsu, Japan) with a pixel size of 6.5 μm and a peak quantum efficiency of 80%. The x-y pixel size is 107.5 nm, and the z-step size is 200 nm. A stack of 41 optical planes (0.2 μm step) was acquired consecutively in 5 channels (652 nm, 573 nm, 759 nm, 495 nm, 409 nm) using Nikon Elements software (Nikon Corporation, Japan).

### 2.7. Imaging Analysis

We have created a modularized computational pipeline in Python, ARLIN (analysis of RNA localization in neurons), to analyze the specific subcellular distribution of single mRNAs in neurons. We provide a detailed description of the program and a “User’s Manual” in [Code information (Appendix A), User’s Manual (Appendix A), https://github.com/VeraUgalde/ARLIN (accessed on 20 May 2022)]. Briefly, ARLIN has two parts, Part 1 and Part 2. Part 1 is a segmentation tool to annotate the different subcellular compartments of the neurons, soma, nucleus, and dendrites. It creates prints from two-dimensional projections (2D) of z-stack images obtained in Fiji ImageJ (NIH) and annotated using image editing software (Preview on a Mac or Paint on a PC). The prints are used as outlines to quantify the number of single mRNAs and their precise localization using FISHQuant [46]. Part 2 has four modules that use the information obtained in Part 1 and FISHQuant to obtain statistics about the density of mRNAs in soma and dendrites, distribution of mRNAs in dendrites, localization in dendritic spines, and degree of mRNA colocalization.

## 3. Results

### 3.1. A Computational Pipeline to Study the Subcellular Localization of eEF1A1 and eEF1A2 mRNAs through Neuronal Development by Quantitative Fluorescence Microscopy

Differences in the expression and subcellular localization of *eEF1A1* and *eEF1A2* mRNAs during hippocampal neuronal development were analyzed by smFISH in primary hippocampal neurons cultured ex vivo [51]. Primary hippocampal neurons isolated from postnatal day 0 or 1 (P0 or P1) mice are optimal for studying neurodevelopment because neurons re-synchronize in vitro and fully differentiate into adult hippocampal neurons [7] (Figure 1A–C). They express key phenotypic features of neurodevelopmental changes on specific days of the in vitro culture (DIV), axon outgrowth (DIV1-3), dendritic branching (DIV3-14), spine formation (DIV9-14), and synaptogenesis (DIV14-21). At DIV14, the number of synapses increases, and the neuronal network is more complex, and, at DIV21, neurons are considered fully mature. Thus, this pipeline enables us to relate *eEF1A1* and *A2* mRNA localization to all aspects of dendritogenesis and postsynapse formation.

We identified the stage of neurodevelopmental progression by immunofluorescence (IF) to detect axons by the specific localization of the Tau protein, dendrites by the localization of the microtubule-associated protein 2 (Map2), and dendritic spines by the expression of the Postsynaptic Density 95 (PSD95) (Figure 1C). We started detecting PSD95 staining over dendritic shafts at DIV9, which correlated with the presence of few interactions between TAU and MAP2 staining (Figure 1C, DIV9). To compare the subcellular distribution of *eEF1A1* and *eEF1A2* mRNAs over neurodevelopment, we designed specific smFISH probes that distinguish *eEF1A1* and *eEF1A2* mRNA sequences, and we combined the staining of each of these mRNAs with double IF to detect simultaneously MAP2 and TAU or MAP2 and PSD95. Co-staining with MAP2 and TAU helps us distinguish a short dendrite from the axon as MAP2 is also present in the initial axon segment (Figure 1C, DIV5). Additionally, during the early phase of neuronal differentiation (DIV1), one neurite is selected to become the axon, most often the longest neurite [52,53]. We only found a few *eEF1A1* and *eEF1A2* mRNAs localizing in axons over neurodevelopment (between four to eight per field of view), and, therefore, we focused our analysis on the dendrites.

To collect statistics on the localization of mRNAs in smFISH images of neurons, we have developed a computational pipeline in Python, ARLIN (analysis of RNA localization in neurons) (Code information (Appendix A), User Manual (Appendix A), https://github.com/VeraUgalde/ARLIN (accessed on 20 May 2022)) (Figure 1D). The program is broken into two parts. The first part, Part 1, is a semi-automatic tool for segmenting the cellular compartments of the neuron, the somas, nuclei, and dendrites. Each segmentation is stored as a binary image, which we refer to as a print. Additionally, Part 1 generates skeletons for each dendrite, a 2D binary image of a single white line representing the midline of a print for a dendrite (Figure 1E). After running Part 1, FISHQuant is used to identify the coordinates of mRNAs (x, y, and z) and postsynapses in the smFISH images [46]. Then, the second part of the program, Part 2, uses both the spot coordinate (x and y) obtained from FISHQuant along with the prints and skeletons obtained from Part 1 to collect statistics on the localization of mRNA. Part 2 enables us to investigate the following: (1) the density of mRNA in cellular compartments, (2) the distribution of mRNA along the skeleton of dendrites, (3) the degree of colocalization between two mRNAs, and (4) the degree of localization of mRNAs at dendritic synapses.

### 3.2. eEF1A1 and eEF1A2 Transcription and mRNA Somatic Abundance Are Differentially Regulated during Neurodevelopment

To compare the somatic expression of *eEF1A1* and *eEF1A2* mRNAs and their transcriptional induction, we undertook parallel smFISH experiments using independent neuronal culture dishes. At DIV1, the nucleus mainly occupies the somatic compartment of cultured neurons, and the accumulation of *eEF1A1* mRNAs was prominent in this small somatic area, with an average of one hundred and forty molecules per soma. On the contrary, only half of the DIV1 neurons expressed *eEF1A2* mRNAs, with an average of four molecules per soma (Figure 2A–C). Following DIV1, all the neurons increased the expression of both *eEF1A1* and *eEF1A2*. This result strongly indicates that hippocampal neurons, similar to motor neurons, only expressed *eEF1A1* during embryonic development and coexpressed both paralogue genes during postnatal development [10,11].

The somatic expression and transcriptional induction of *eEF1A1* are regulated differently from *eEF1A2*. We found that the most significant increase in the number of *eEF1A1* mRNAs per soma occurred from DIV3 to DIV5. DIV5 also endowed the highest transcriptional activation of the *eEF1A1* gene, which remained lower but constant at subsequent DIVs until DIV21 (Figure 2B,E). Although we quantified fluctuations in the number of *eEF1A1* mRNAs per soma from DIV5 to DIV21, the extent of somatic *eEF1A1* mRNAs remained steady (Figure 2B). On the contrary, the quantity of *eEF1A2* mRNAs per soma significantly augmented at each step of neurodevelopment until DIV14 (Figure 2C). This increase was supported by a continuous significant rise in the transcription of the *eEF1A2* gene (Figure 2D). To rule out the possibility of an inaccuracy in the quantification of mRNA molecules derived from the increased size of the soma over neurodevelopment, we adjusted our quantifications to mRNAs per pixel of soma and obtained the same results (Appendix A). Thus, hippocampal neurons, similar to motor neurons, increased the expression of eEF1A2 as neurodevelopment progressed. However, the expression of eEF1A1 was not repressed at DIV21 in mature neurons. Since hippocampal neurons cannot be maintained in culture for longer than 28 days, we analyzed whether the transition in the expression from eEF1A1 to eEF1A2 occurs in the mouse hippocampus and whether the expression of *eEF1A1* mRNA persisted in old hippocampal neurons. To this end, we analyzed tissue samples obtained from 6- and 90-week-old C57BL/6J mice by smFISH. Interestingly, *eEF1A1* mRNA is very abundant in the hippocampus, cell body, and processes of 6-week-old mice, and *eEF1A2* transcription sites and a few single mRNAs were also detected in these tissues (Appendix A). In aged mice (90-week-old), the coexpression of both *eEF1A1* and *eEF1A2* persisted in all parts of the hippocampus, dentate gyrus (DG), CA1, CA2, and CA3 neurons. We detected single *eEF1A1* mRNAs in both the soma and the neurite compartments, and *eEF1A1* were more abundant than *eEF1A2* mRNAs in the neurite fraction (Figure 1E and Appendix A). Thus, the coexpression of both eEF1A paralogues observed in the cultured primary neurons at DIV21 reflects the pattern of expression existing in mature and old hippocampal neurons.

### 3.3. The Dendritic Concentration of eEF1A1 and eEF1A2 Increase over Neurodevelopment

We observed single *eEF1A1* mRNAs over the dendritic shaft starting at DIV1 (Figure 2A and Figure 3A). The quantity of eEF1A1 consistently increased over development, except at DIV9, from an average of six mRNAs at DIV1 to thirty-eight at DIV21 (Figure 3B). We quantified the most significant increase in *eEF1A1* mRNAs per dendrite at DIVs 14 and 21 at the time of synaptogenesis, which indicates that *eEF1A1* mRNA localization in dendrites is needed for both neurodevelopmental and fully differentiated hippocampal neurons. Similar to *eEF1A1*, the quantity of *eEF1A2* mRNAs in neurites significantly increased over development until DIV14, from an average of two mRNAs at DIV3 to eight mRNAs at DIV21 (Figure 3C). These quantifications indicate that the average number of *eEF1A2* mRNAs per dendrite was consistently lower than *eEF1A1* mRNAs. For example, dendrites of DIV14 neurons have an average of thirty *eEF1A1* and ten eEF1A2 mRNAs. This result suggests that eEF1A1 might have specific functions suited for developmental and dendritic requirements of mature neurons that are not fulfilled by eEF1A2.

We observed extensive variability in the number of mRNAs per dendrite for both eEF1As; while some dendrites have almost no mRNAs, others were crowded with mRNAs (Figure 2A and Figure 3B,C). We used the module “distance of mRNA to the soma” of ARLIN to calculate the distance that each mRNA has traveled in the dendrite from the soma to study the distribution of mRNAs over the dendritic shaft (Appendix A). The distances were grouped by one of the following bins: 0–25 µm, 25–50 µm, 50–75 µm, 75–100 µm, 100–125 µm, 125–150 µm, or >150 µm. We found that, despite the variability in the number of mRNAs per dendrite, the distribution of *eEF1A1* and *eEF1A2* mRNAs over the dendritic shaft was similar. Most of the *eEF1A1* and *eEF1A2* mRNAs accumulated in the proximal dendrite at the first 50 μm from the soma. As the total number of *eEF1A1* mRNAs in the dendrites increased through neurodevelopment, we observe that their number also augmented in the distal part of the dendrite (Figure 3D). The level of dendritic *eEF1A2* mRNAs is low, but mRNAs were transported to the distal dendrite as we identified a few dendrites bearing *eEF1A2* mRNAs in the 100–125 and 125–150 µm segments (Figure 3E).

Interestingly, we spotted several *eEF1A1* mRNAs towards the tip of growing dendrites at DIV1 and in areas of dendritic branching from DIV3 until DIV9 (Figure 3A (white arrows)). Thus, *eEF1A1* mRNA might be locally translated to provide a local source of eEF1A1 to sustain the ongoing protein synthesis and cytoskeleton reorganization and support dendritic outgrowth and branching at the early stages of neurodevelopment (Figure 3A (white arrows)). Likewise, at DIV9, some *eEF1A1* mRNAs localized at the edge of the dendritic Map2 signal, suggesting that they might sustain the outgrowth of the dendritic spines (Figure 3A (yellow arrows)). Hence, our data indicate that eEF1A1 might have several functions related to the morphological changes that characterize neurodevelopment.

### 3.4. eEF1A1 and eEF1A2 mRNAs Are Transported Independently and Have a Distinct Localization at Dendritic Spines

Postsynapses (dendritic spines) start being formed at DIV9. In order to study the localization of *eEF1A1* and *eEF1A2* mRNA at synapses, we combined a double IF to detect the dendritic shaft with the Map2 antibody and the dendritic spines with the PSD95 antibody, with smFISH to identify either *eEF1A1* or *eEF1A2* mRNAs (Figure 4A). We quantified the degree of *eEF1A1* or *eEF1A2* mRNA synaptic “localization” using the “synapse statistic” module of ARLIN, which detects the number of mRNAs within a given distance to a dendritic spine (Appendix A). Dendritic spines are characterized by a high protein concentration sustained by local mRNA translation. Further, mRNAs translated locally either localize inside or at the base of the dendritic spine [54,55,56,57]; thus, we considered mRNAs within a 500 nm radius from the centroid of the PSD95-IF signal as “mRNAs localizing in the spines.” To determine whether the localization of mRNAs in dendritic spines is either regulated or random, we quantified the number of spines endowed with at least one mRNA, and then we simulated 50 times the localization of the mRNAs on the dendritic shaft, and we compared the percent of spines with at least one mRNA in the real and simulated data (Figure 4B).

We quantified a higher percent of spines with *eEF1A1* than *eEF1A2* mRNAs in DIVs 9, 14, and 21 in both the real and simulated data due to the higher level of *eEF1A1* mRNA trafficking in dendrites. The number of synapses endowed with at least one *eEF1A1* mRNA increased over neurodevelopment and was higher than the simulated data at DIVs 14 and 21, with ~12% of spines containing at least one mRNA. The highest difference between the real and simulated data was at DIV14, during synapsis formation (synaptogenesis) in neurons, wherein synapses were shown to attract four *eEF1A1* mRNAs (Figure 4B). These data suggest that localizing mRNAs might support the local synthesis of eEF1A1 in order to sustain synaptogenesis. On the contrary, the real levels of *eEF1A2* mRNA were not higher than the simulated data, suggesting that the localization of *eEF1A2* mRNAs in dendritic spines occurs randomly and might not be necessary during developmental processes (Figure 4B).

We next investigated whether *eEF1A1* or *eEF1A2* mRNA molecules travel together in neuronal granules, which has been suggested for other dendritic mRNAs [58,59]. We rationalized that, if mRNAs travel together in a granule, they should be in close proximity. Thus, the mRNAs should have a short intermolecular distance. We developed the “mRNA colocalization” module in ARLIN to measure and plot the distance between an mRNA and the closest mRNA molecule (Figure 4C,D, Appendix A). This module also simulated random localizations of mRNAs and plotted their intermolecular distances (Figure 4C,D). We did not obtain any significant differences between the real and simulated data for both *eEF1A1* and *eEF1A2* mRNAs, pointing to the individual trafficking of *eEF1A1* and *eEF1A2* mRNAs along the dendritic shaft.

### 3.5. eEF1A1 and eEF1A2 mRNAs Have a Different Subcellular Distribution That Changes over Development

To relate subcellular specific changes in the expression of *eEF1A1* and *eEF1A2* mRNAs over neurodevelopment, we calculated the fold change (FC) in the expression of *eEF1A2* versus *eEF1A1* in the somatic and dendritic compartments separately. We found that, while *eEF1A1* expression remains prevalent in developing and mature hippocampal neurons in both compartments, the levels of *eEF1A2* mRNA in the soma progressively augment to almost equal levels as those of the *eEF1A1* mRNA by DIV21 (Figure 5A). However, the fold increase of *eEF1A2* versus *eEF1A1* mRNA expression behaved differently in dendrites. While *eEF1A2* mRNA increased until DIV9, *eEF1A1* mRNA levels took over again during synaptogenesis, further implying a role for eEF1A1 in supporting synapse formation.

Furthermore, our data revealed that few *eEF1A1* and *eEF1A2* mRNAs localize in axons over neurodevelopment (between four and eight per field of view). Hence, we calculated whether *eEF1A1* and *eEF1A2* mRNAs have a preferential somatic or dendritic localization using the mRNA density values (mRNAs per pixel of soma or dendrite obtained with ARLIN). In both cases, the mRNA localization is preferentially somatic (Figure 5B,C). However, in the case of *eEF1A1* mRNA, its localization was less somatic at DIVs 14 and 21 than at early developmental stages (DIV1-9), while the opposite was found for *eEF1A2* mRNA. Thus, adult hippocampal neurons maintain *eEF1A2* mRNAs in the soma while exporting *eEF1A1* mRNAs to the dendrites. We validated the smFISH data by RT-qPCR to quantify the expression of *eEF1A1* and *eEF1A2* mRNAs in the somas and neurites of DIV21 neurons. We found that the expression of *eEF1A1* and *eEF1A2* is preferentially somatic and that eEF1A1 is the predominant variant in both compartments (Figure 5D,E). These results indicate that somatic and dendrite compartments of mature hippocampal neurons might have different functional and morphological requirements that involve the preferential expression of an eEF1A paralogue over the other.

## 4. Discussion

Neurons have evolved a decentralized gene expression system to localize mRNAs at different neuronal compartments and regulate their translation. Indeed, mRNA localization coupled with local translation is the most efficient strategy for neurons to target proteins to dendritic spines [54,55]. This is especially relevant for proteins needed in high quantities at long distances from the soma because the mRNAs provide an on-demand local source of proteins through the regulation of protein synthesis. Accordingly, mRNAs encoding proteins implicated in translation, such as eEF1A, exit the soma and are locally translated [3,59,60,61,62]. Using the mRNA as a proxy for protein expression and localization in neurons, we used smFISH to study changes in the expression and subcellular distribution of *eEF1A1* and *eEF1A2* transcripts of individual neurons during hippocampal neuronal development and in fully differentiated hippocampal neurons. Due to the lack of established automatic techniques to segment widefield smFISH images in neurons, we developed ARLIN, a semi-automatic and compartmentalized computational framework, to analyze the expression and degree of the subcellular localization of mRNAs in neurons. Neurons have a complex morphology compared to other cell types, and, thus, classic techniques, such as the watershed method, and more advanced techniques, such as U-Net, which introduced a novel deep-learning architecture and training strategy for cell segmentation, are not well adapted for neurons [63]. Extensive work has been carried out to segment neurons from 3D optical confocal microscope images as an intermediate step of neuron tracing [64,65]. Despite such advances, methods to segment microscope images of neurons have been limited to the segmentation of only the somatic compartment [66]. Though our pipeline still requires manual annotations, our algorithm for inconsistent signals makes ARLIN robust to imprecise annotations and noisy images. The main advantage of ARLIN is that it subtracts spatial information from the intact morphology of neurons, allowing us to compare large datasets automatically. ARLIN uses the output of the mRNA coordinates obtained from the smFISH analysis by FISHQquant [46] to provide accurate quantifications of (1) the density of mRNAs in soma and dendrites, (2) the distance that mRNAs travel from the soma into dendrites, (3) the spatial relationship among mRNAs encoding from the same or different proteins to establish principles on their transport over dendrites, and (4) their degree of colocalization with dendritic spines to investigate the regulation of their distribution under stimuli, such as neurodevelopmental clues.

Using ARLIN, we described precise changes in the subcellular localization of *eEF1A1* and *eEF1A2* mRNAs over neurodevelopment. As expected, *eEF1A2* mRNA is not expressed in undifferentiated neurons, and its expression becomes elevated over neurodevelopment [9,10]. Further, eEF1A2 has an anti-apoptotic function [67] and a capacity for co-translational targeting of nascent peptides to proteasome degradation under stress conditions [68], which would support homeostasis throughout the extended lifetime of functional neurons. We also discovered that *eEF1A1* and *eEF1A2* mRNAs exhibited different subcellular distributions that seem to be coordinated with specific neuronal morphology changes. Overall, the dendritic levels of *eEF1A1* were higher than *eEF1A2* mRNAs and showed specific localization patterns. Specifically, *eEF1A1* mRNAs were present at the tip of the extending neurites at DIV1 during neurite outgrowth. Their expression at the endpoints may be involved in dendritic branching at DIV9 when dendritogenesis occurs. Furthermore, *eEF1A1* mRNAs were expressed in dendritic spines, suggesting their involvement in synaptogenesis. Thus, eEF1A1 but not eEF1A2 might coordinate the extensive remodeling of the neuronal cytoskeleton with the vast protein synthesis [7]. In fact, eEF1A1 has a higher affinity for GTP and the guanine-nucleotide exchange factor complex eEF1B than eEF1A2 [8], which could lead to a faster processivity and affect the fate and functioning of the local proteome [69,70,71]. Furthermore, eEF1A1 brings mRNAs in contact with the cytoskeleton and modifies the actin cytoskeleton [72]. Therefore, it can facilitate the rearrangements required for neuronal differentiation into a complex morphology and establish protein networks at synapses [73,74]. Once differentiated, eEF1A2 would support a longstanding architecture and interneuron network through binding and bundling the actin cytoskeleton [23]. Accordingly, it was recently demonstrated that eEF1A2 balances actin bundling and local translation elongation in the dendritic spines of hippocampal neurons upon synaptic stimulation through the phosphorylation of its specific Serine 358 amino acid, which is substituted by an Alanine in eEF1A1 [19]. This study suggests that the functions of eEF1A2 are modulated by the neuron to adapt to stimuli. Given that *eEF1A1* is more prominent in dendrites than *eEF1A2* mRNA (Figure 5) [75], we expect future studies to uncover whether specific eEF1A1 amino acids are important for neuronal activity. In addition, differences in the posttranscriptional regulation of eEF1A1 and eEF1A2 can allow them to share and complement their functions under different conditions. These two variants have different 5′and 3′ untranslated regions; the 3′UTR of *eEF1A2* is twenty times longer than *eEF1A1* mRNA, which advocates for a complex regulation by *cis*-regulatory elements and RNA binding protein partners. Although both 5′UTRs are polypyrimidine-rich, *eEF1A1* mRNA has been long established as a 5′TOP mRNA, and, thus, its translation can be rapidly adapted to environmental and internal stimuli [76,77], which will also ensure its rapid translation [59]. Thus, a source of eEF1A1 proteins might act to sustain the ongoing translation of hundreds of dendritic mRNAs. We foresee that investigating the regulation of translation of these mRNAs with novel single-molecule fluorescence reporters, such as the SunTag system [78,79], will provide precise quantifications on the regulation of each eEF1A variant translation and the means to relate protein synthesis with mRNA distribution.

It was unexpected to find that mature hippocampal neurons coexpress *eEF1A1* and *eEF1A2* mRNAs in in vitro cultured neurons and tissue samples obtained from 90-week-old mice (Figure 2, Figure 3, Figure 5 and Appendix A). It is well accepted that the expression of these two paralogue genes is mutually exclusive in adult cells, and only cancer cells express them simultaneously [9,18,80,81]. We should consider that *eEF1A1* mRNAs might be translationally repressed, which would explain the discrepancy between our results and those of previous studies using antibodies. Alternatively, hippocampal neurons could be the exception to this rule. We think that hippocampal neurons coexpress eEF1A1 and eEF1A2 proteins because we detect active transcription sites for the *eEF1A1* locus, even in tissue samples (Appendix A), which advocates for continuous synthesis of the *eEF1A1* mRNA. Moreover, the *eEF1A1* mRNA is translocated from the soma to the dendrites, and its localization becomes more dendritic in mature than in developmental neurons (Figure 5B). This result suggests that the subcellular distribution of *eEF1A1* mRNA is well regulated. Identifying the regulatory elements of the *eEF1A1* mRNA that direct its dendritic localization is important to study the role of this specific isoform during hippocampal neuron development and in mature neurons.

Overall, our results point to differences in the switch from eEF1A1 to eEF1A2 expression between motor neurons and hippocampal neurons. We suggest the differences in the architecture and function among neuronal types impose specific functions undertaken by either eEF1A1 or eEF1A2. In the case of hippocampal neurons, the high translational demands of the dendritic compartment might be better suited for eEF1A1 and complemented by eEF1A2 [19]. It will be beneficial to use ARLIN to study the intriguing change in the expression of the eEF1A variants in different neuronal types (including motor neurons) upon synaptic stimulation and under pathological conditions. Likewise, ARLIN is helpful in investigating the role of cis- and trans-regulators in orchestrating the complex distribution of eEF1A variants and other mRNAs over neuronal morphology and how mutations lead to the mislocalization of mRNAs and neuropathological diseases.

## Figures and Tables

**Figure 1 cells-11-01877-f001:**
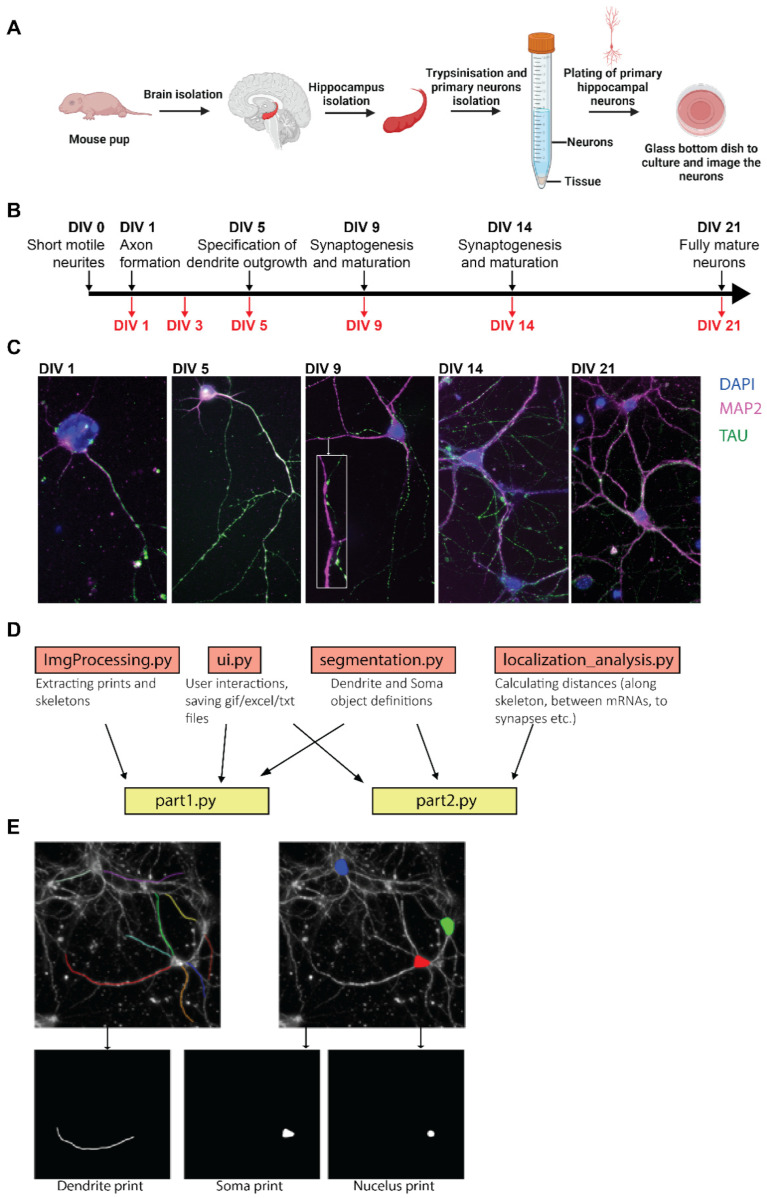
Pipeline to generate and analyze the subcellular localization of mRNAs in primary hippocampal neurons. (**A**) Schematic of the protocol to isolate primary hippocampal neurons from P0–P1 pups. (**B**) Black arrow indicates the timeline of hippocampal neuron development in culture. Red arrows indicate the days in in vitro culture (DIV) at which neurons were fixed and the specific neurodevelopment characteristic of each DIV. (**C**) Representative fluorescence microscopy images obtained from neurons fixed at the indicated days. Nucleus stained with DAPI (blue) and immunofluorescence (IF) to detect Map2 in dendrites (magenta) and Tau in axons (green). Scale bar is 10 mm. (**D**) ARLIN computational program layout, the function of each module, and their dependencies. ARLIN is composed of 6 modules, two of which are executable, part1.py and part2.py. Both Part 1 and Part 2 draw from ui.py and segmentation.py. (**E**) Schematic of input and output of part1.py. Images of annotated dendrites and somas are used to generate prints and skeletons.

**Figure 2 cells-11-01877-f002:**
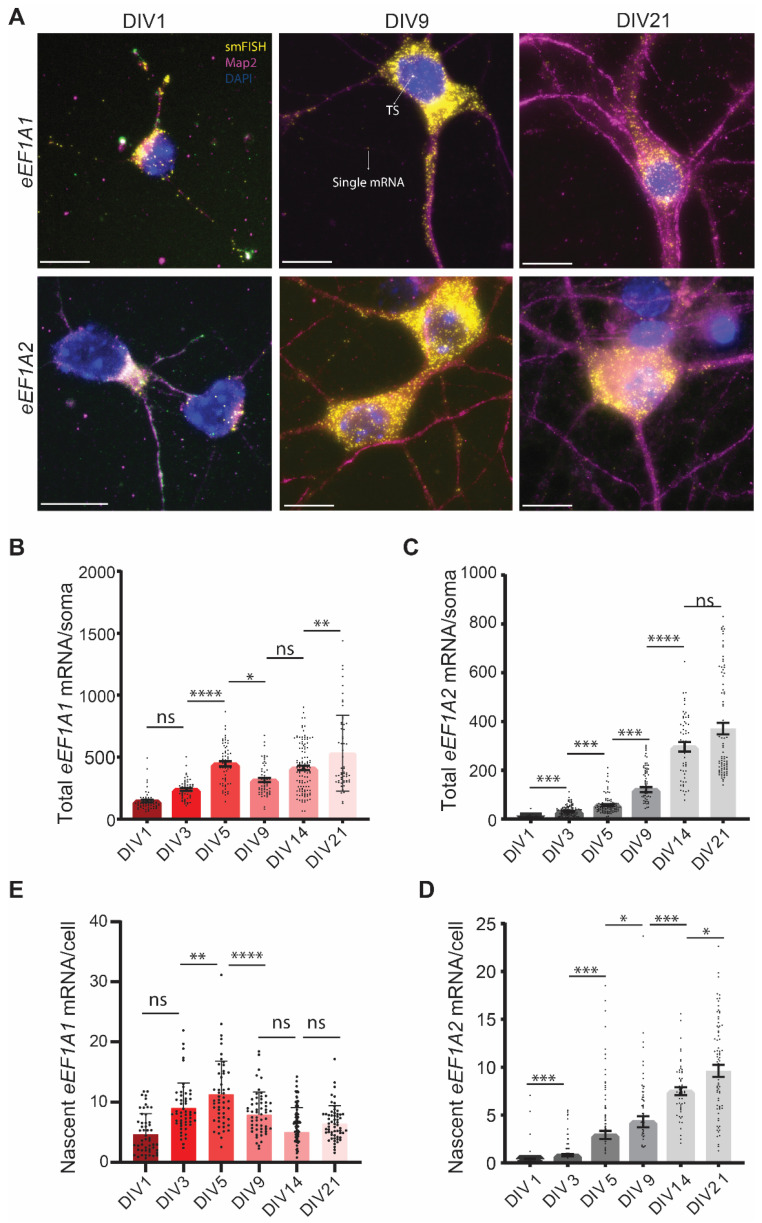
Expression of *eEF1A1* and *eEF1A2* mRNAs in the soma through neurodevelopment. (**A**) smFISH-IF to detect *eEF1A1* or *eEF1A2* mRNAs in soma and dendrites. Representative smFISH-IF at the indicated DIVs. Nucleus stained with DAPI (blue), eEF1A1 and eEF1A2 smFISH colored in yellow, and dendrites stained by IF to detect Map2 indicated in magenta. TS indicates the signal of a transcription site and the arrow in DIV9 indicates the signal from a representative single mRNA. Scale bar is 10 mm. (**B**,**C**) Number of mature *eEF1A1* (**B**) and *eEF1A2* (**C**) mRNAs per soma through neurodevelopment. (**D**,**E**) Number of nascent *eEF1A1* (**D**) and *eEF1A2* (**E**) transcripts per nucleus through neurodevelopment. Total number of neurons analyzed in four experiments, eEF1A1 N = 81–118, eEF1A2 N = 87–172. Statistical analysis to compare a specific DIV with the next developmental time point was conducted with one-way ANOVA (*p*-value ns > 0.05, * < 0.05, ** < 0.01, *** < 0.001, **** < 0.0001).

**Figure 3 cells-11-01877-f003:**
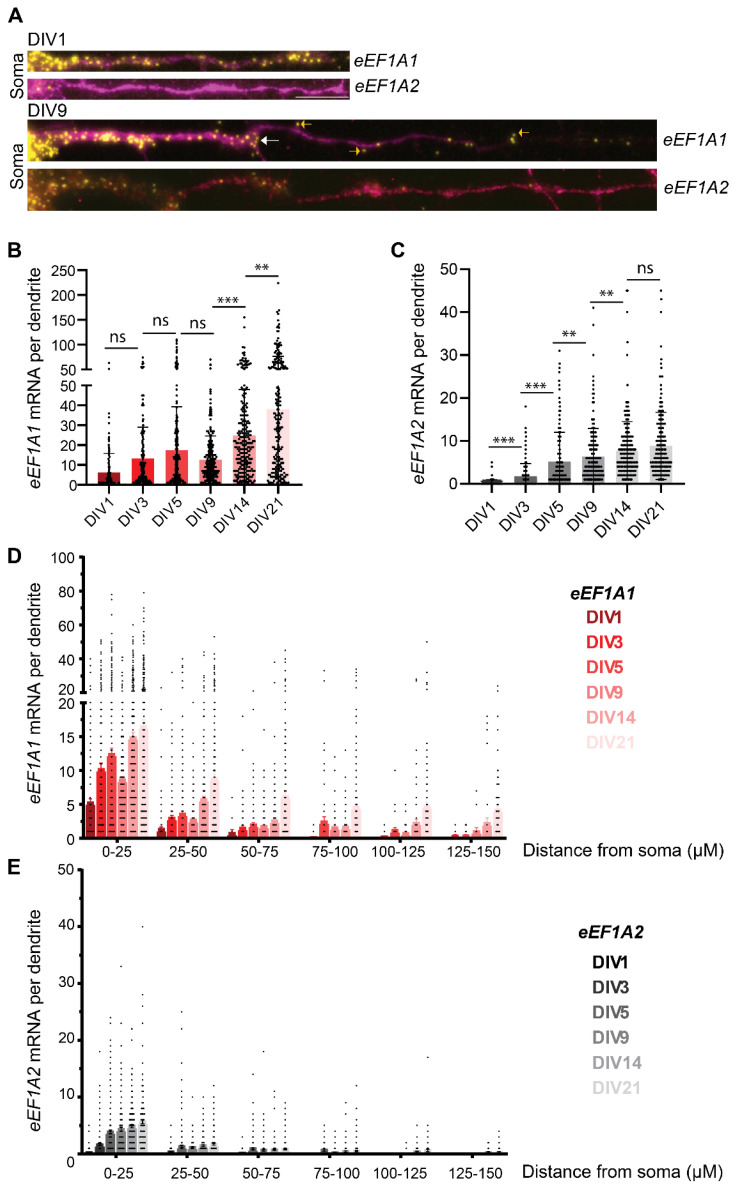
*eEF1A1* and *eEF1A2* mRNA localization in the dendrites through neurodevelopment. (**A**) smFISH–IF to detect *eEF1A1* or *eEF1A2* mRNAs in dendrites. Representative smFISH-IF at the indicated DIVs. eEF1A1 and eEF1A2 smFISH colored in yellow, and dendrites stained by IF to detect Map2 indicated in magenta. Scale bar 5 mM. White arrows indicate mRNAs at the dendritic branching site and yellow arrows point to mRNAs outside the dendritic IF staining. (**B**,**C**) Number of *eEF1A1* (**B**) and *eEF1A2* (**C**) mRNAs per dendrite. Number of dendrites, eEF1A1 N = 147–238, eEF1A2 N = 115–241. Statistical analysis to compare a specific DIV with the next developmental time point was conducted with one-way ANOVA (*p*-value ns > 0.05, * < 0.05, ** < 0.01, *** < 0.001). (**D**,**E**) distribution of *eEF1A1* (**D**) and *eEF1A2* (**E**) mRNA along the dendrites through neurodevelopment. Bin size: 25 µm of dendrite segmented as a distance from the soma. Number of dendrites, eEF1A1 N = 147–238, eEF1A2 N = 115–241.

**Figure 4 cells-11-01877-f004:**
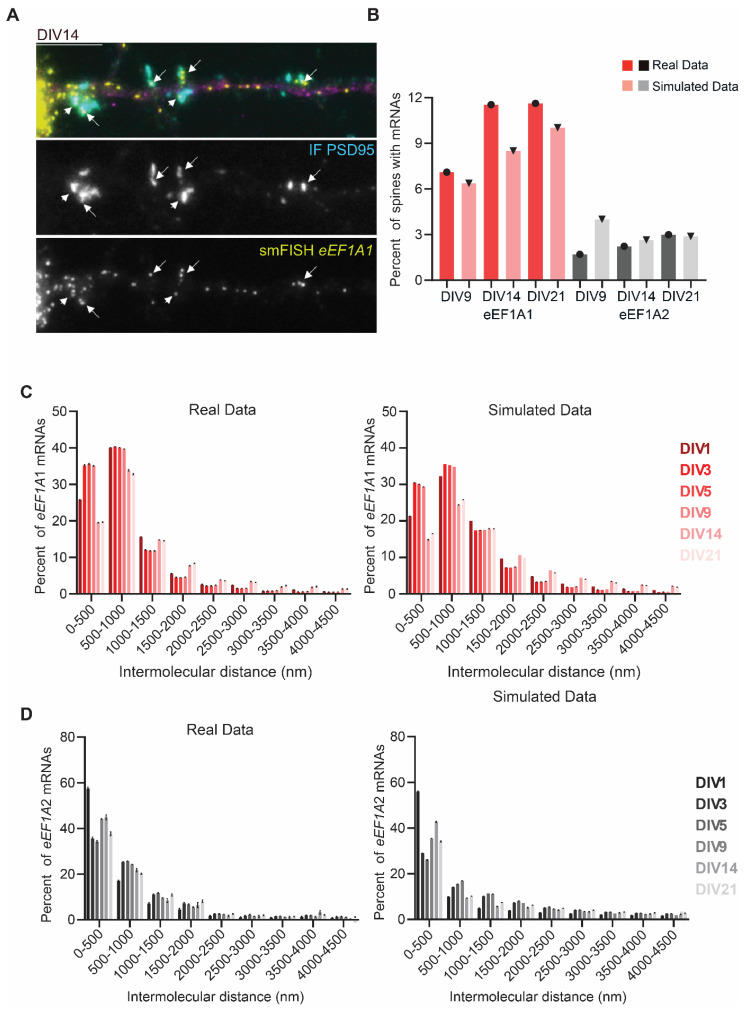
*eEF1A1* and *eEF1A2* mRNA colocalization in dendrites and localization in dendritic spines. (**A**) Representative smFISH–IF to detect *eEF1A1* mRNAs in dendrites and synapses at DIV14. Top panel shows the merge signal of dendritic staining with Map2 (magenta), dendritic spine staining with PSD95 (cyan), and *eEF1A1* mRNA detection by smFISH (yellow). Middle panel indicates the IF to detect PSD95 and low panes the smFISH to detect *eEF1A1* mRNA. Arrows indicate synaptic densities occupied with at least one *eEF1A1* mRNA. Scale bar is 10 mm. (**B**) Average percentage of synapses with 1 to 5 mRNAs within 500 nm from the centroid of the IF signal at DIV9, 14, and 21 obtained from the real data eEF1A1 N = 105–145, eEF1A2 N = 61–131, and from simulated data by running 50 simulations of random localization of the exact number of experimental RNAs with module 3 of ARLIN. (**C**,**D**) Evaluating mRNA colocalization of *eEF1A1* and *eEF1A2* during neuronal development. Percent of mRNA “colocalization” of absolute (real) *eEF1A1* mRNA to the closest *eEF1A1* mRNA, and simulated *eEF1A1* to the closest *eEF1A1* mRNA (**C**), and of real *eEF1A2* mRNA to the closest *eEF1A2* mRNA, and *eEF1A2* to the closest *eEF1A2* mRNA (**D**). Intermolecular distance for mRNA colocalization calculated for 4500 nm and plotted every 500 nm distance using ARLIN. Simulated data were conducted a total of 25 times using module 3 of ARLIN. Total number of mRNA for actual eEF1A1: DIV1 N = 3912, DIV3 N = 20,001, DIV5 N = 22,702, DIV9 N = 23,762, DIV14 N = 4803, DIV 21 N = 10,340; simulated eEF1A1: DIV1 N = 98,934, DIV3 N = 501,970, DIV5 N = 569,099, DIV9 N = 595,171, DIV14 N = 119,624, DIV 21 N = 260,536; actual eEF1A2: DIV1 N = 5659, DIV3 N = 5112, DIV5 N = 6155, DIV9 N = 17,099, DIV14 N = 2659, DIV 21 N = 3649; simulated eEF1A2: DIV1 N = 136,913, DIV3 N = 122,106, DIV5 N = 147,579, DIV9 N = 416,122, DIV14 N = 64,906, DIV 21 N = 86,398. SEM bars. Kolmogorov–Smirnov *t*-Test *p* > 0.05 ns.

**Figure 5 cells-11-01877-f005:**
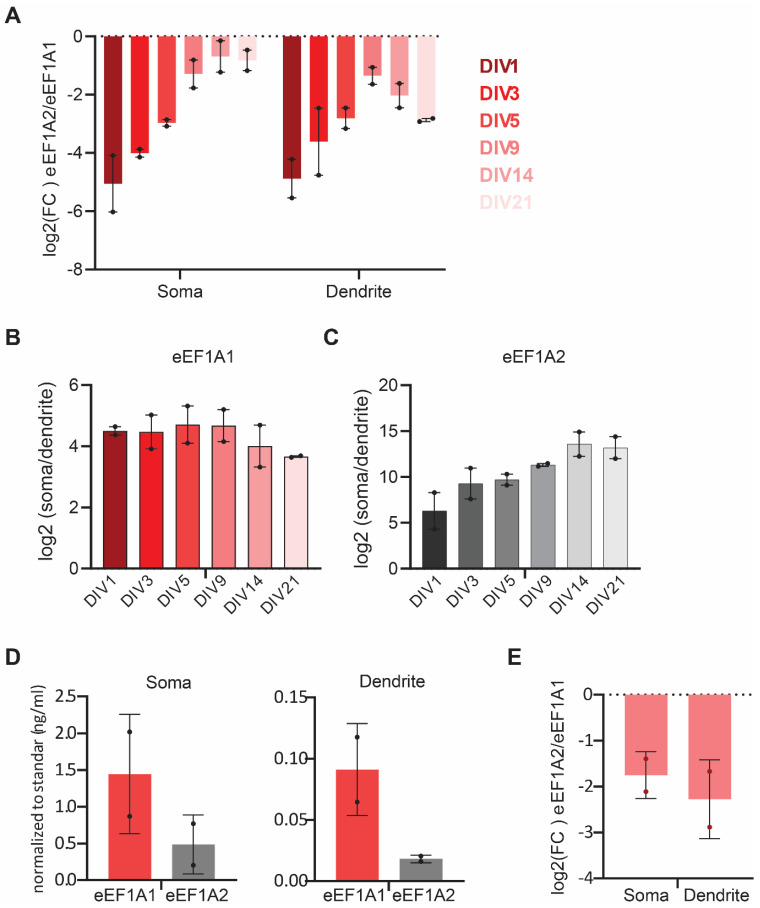
Relation of *eEF1A1* and *eEF1A2* mRNA subcellular localization over neurodevelopment. (**A**) Fold change in the expression of eEF1A2 vs. eEF1A1. Plot of the log2 of the ratio between *eEF1A2* and *eEF1A1* mRNA molecule density at the indicated DIVs in somas and in dendrites from experiments quantified in Figure 2 and Figure 3. (**B**,**C**) Fold change in the somatic versus the dendritic number of eEF1A1 (**B**) and eEF1A2 (**C**) mRNAs at the indicated DIVs. (**D**) RT-QPCR to detect *eEF1A1* and *eEF1A2* mRNAs from total RNA extracted from the soma or neurite fractions of DIV21 neurons was normalized to a standard curve obtained from serial dilutions (50, 5, 0.5, and 0.05 ng/mL) of total cDNA. Average and SD of two biological replicas. (**E**) Plot of the log2 of the ratio between *eEF1A2* and *eEF1A1* mRNA relative concentrations in somas and dendrites obtained from (**D**).

## Data Availability

The code is available in a public repository on GitHub. (https://github.com/VeraUgalde/ARLIN (accessed on 11 May 2022)).

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
