# Peer review of "Analysis of the Expression and Subcellular Distribution of eEF1A1 and eEF1A2 mRNAs during Neurodevelopment"

_cells, 2022, doi:10.3390/cells11121877_

Round 1

Reviewer 1 Report

This is a really nice paper that will be a useful addition to the field. A novel open-source computational framework for analysis of transcripts in neurons is described and presented; this is an excellent resource that will be of general interest. The data are really well presented with beautiful imaging of the cultured cells.

Major comments:

The wording throughout should really be slightly more balanced- the authors present a strong argument for RNA as a surrogate for protein level expression but given the level and sophistication of post-transcriptional regulation in neurons it would help to be more nuanced. Similarly, all the work bar the one supplementary figure (see further comment below) was carried out in cultured cells. It is highly likely that this will influence the relative expression of eEF1A variants and this should at least be acknowledged. It would have been really helpful to have looked at RNA levels of the two variants in motor neurons, where the protein level switch seems to be complete, though obviously this is not a requirement for the current paper. Similarly, the use of “adult” to describe cells at DIV21 is rather misleading and the paper would be improved by not over-simplifying. An example is line 324 “which indicates that eEF1A1 mRNA localization in dendrites is needed for both neurodevelopmental and adult hippocampal neurons”- needed is a really strong statement.

The only results suggesting that the cultured cells reflect the situation in adult brain are in the supplementary figure, showing one high power image with only a few neuronal cell bodies (different numbers for eEF1A1 and eEF1A2). This is an important piece of evidence and comparative images showing the whole hippocampus should be included.

Minor comments:

There is a reference to “method paper” where there should be a proper reference (line 117).

It reads slightly oddly, when the introduction is so heavily focused on eEF1A, to have such a heavy emphasis on the computational pipeline at the start of the discussion, though this is not a big problem.

Reviewer 2 Report

In this manuscript, Wefers et al. analyze the subcellular distribution of eEF1A1 and eEF1A2 mRNAs in neuronal cell development. This work is comprehensive in many aspects, especially, it fills the gap to determine the dynamics of two eEF1A1 mRNA isoforms during a series of development stages, rather than a single-time-point snapshot of expression profiles. Moreover, this work presents a user-friendly software called ARLIN that can be used for image data analysis and segmentation. The source codes are free to the public, which allows the reproducibility of the study.

The authors found the co-expression of eEF1A1 and eEF1A2 isoforms from cultured neurons, which is unexpected because they were believed to be mutually exclusive. I wonder whether this discovery can be supported by further evidence, such as RNA-Seq and qPCR experiments? What’s the biological significance to have these two isoforms expressed together? I look forward to more discussions.

Overall, I support the publication of this work under minor revision.
